# An Exploratory Application of Multilayer Networks and Pathway Analysis in Pharmacogenomics

**DOI:** 10.3390/genes14101915

**Published:** 2023-10-07

**Authors:** Marianna Milano, Giuseppe Agapito, Mario Cannataro

**Affiliations:** 1Department of Experimental and Clinical Medicine, University Magna Græcia of Catanzaro, 88100 Catanzaro, Italy; 2Data Analytics Research Center, University Magna Græcia of Catanzaro, 88100 Catanzaro, Italy; agapito@unicz.it (G.A.); cannataro@unicz.it (M.C.); 3Department of Law, Economics and Social Sciences, University Magna Græcia of Catanzaro, 88100 Catanzaro, Italy; 4Department of Medical and Surgical Sciences, University Magna Græcia of Catanzaro, 88100 Catanzaro, Italy

**Keywords:** pharmacogenomics, network analysis, multilayer networks, community detection, pathway enrichment analysis

## Abstract

Over the years, network analysis has become a promising strategy for analysing complex system, i.e., systems composed of a large number of interacting elements. In particular, multilayer networks have emerged as a powerful framework for modelling and analysing complex systems with multiple types of interactions. Network analysis can be applied to pharmacogenomics to gain insights into the interactions between genes, drugs, and diseases. By integrating network analysis techniques with pharmacogenomic data, the goal consists of uncovering complex relationships and identifying key genes to use in pathway enrichment analysis to figure out biological pathways involved in drug response and adverse reactions. In this study, we modelled omics, disease, and drug data together through multilayer network representation. Then, we mined the multilayer network with a community detection algorithm to obtain the top communities. After that, we used the identified list of genes from the communities to perform pathway enrichment analysis (PEA) to figure out the biological function affected by the selected genes. The results show that the genes forming the top community have multiple roles through different pathways.

## 1. Introduction

Network analysis is a branch of network science that deals with the study of complex networks. To investigate complex relationships, network analysis adopts theories and methods typical of several research areas [1]. Networks and network analysis methods are a keystone in computational biology and bioinformatics and are increasingly being used to study biological and clinical data in an integrated way. In detail, network analysis consists of a collection of techniques with a shared methodological perspective, which allows to depiction of relations among entities and to analysis of the structures that emerge from the recurrence of these relations. The basic assumption is that better explanations of different phenomena are yielded by the analysis of the relations among entities. A classical network analysis method is represented by community detection [2]. Community detection is one of the most popular research areas in various complex systems, such as biology, sociology, medicine, and transportation systems [3,4]. The reason for this is that the community structures, defined as groups of nodes that are more densely connected than the rest of the network, represent significant characteristics for understanding the functionalities and organizations of complex systems modelled as a network [2]. It is expected that the communities play significant roles in the structure-function relationship. For example, in biological networks such as protein–protein interaction (PPI) networks, the communities represent proteins involved in a similar function; in neuroscience, the communities detected in brain networks mean regions of interest (ROI) that are active during tasks. In social networks, communities can be groups of friends or colleagues. In the World Wide Web, communities represent web pages sharing the same topic [5]. Thus, the discovery of communities in these systems has become an interesting approach to figuring out how network structure relates to system behaviours. In recent years, network analysis has become an essential tool in pharmacogenomics [6,7]. By providing a powerful framework to model data, network analysis has allowed researchers to analyse and interpret complex interactions between genes, proteins, and drugs in the pharmacogenomics field. This enables uncovering underlying biological mechanisms, identifying potential drug targets and biomarkers, facilitating drug repurposing efforts, and enabling personalized medicine approaches. In particular, network analysis can identify potential drug targets by constructing biological networks that integrate various data sources, such as protein–protein interactions, gene expression data, and pathway information. Furthermore, by analysing the network topology and identifying key nodes or modules, it is possible to pinpoint genes or proteins that play crucial roles in disease pathways or drug responses [8,9,10]. This information can guide the development of targeted therapies. Also, network analysis can aid in the discovery of genetic biomarkers that predict drug response or adverse reactions. By integrating genomic and clinical data, researchers can construct networks that capture the relationships between genetic variations, clinical phenotypes, and drug response [11]. Network-based approaches can identify modules or subnetworks that are highly associated with specific drug responses, enabling the discovery of potential biomarkers for personalized medicine. Also, network analysis can uncover the interconnected pathways and biological processes affected by genetic variations or drug treatments. By mapping genetic variants onto biological networks, pathways that are significantly enriched for these variants can be identified [12]. This knowledge helps in understanding the molecular mechanisms underlying drug response and identifying potential targets for intervention.

By analysing the interactions between drugs, genes, and diseases in a network context, potential off-target effects or repurposed drugs for different indications can be revealed [13]. Finally, network analysis can contribute to personalized medicine approaches by integrating patient-specific genetic and clinical data into networks.

Recently, the need to investigate more complicated frameworks than the classical networks has led to the introduction of a multilayer approach as an extension of graph theory. The reason for this is that many real networks cannot be exhaustively explained with a classical network approach, but need more complex structures [14,15]. The introduction of multilayer networks provides a more comprehensive and realistic representation of complex systems where multiple types of relationships coexist. It allows to analyse and understand the dynamics and behaviour of interconnected entities in a more nuanced manner [16].

Multilayer network analysis enables the study of various properties and phenomena that are not easily captured by traditional network analysis approaches. It allows for the examination of interdependencies, correlations, and patterns that emerge across different layers. This can provide insights into how different layers influence each other, the resilience of the system, the spread of information or diseases, and the identification of key nodes or communities in the network.

Starting from these considerations, in this work, we aim to present an application of network analysis in pharmacogenomics to demonstrate how network analysis methods are able to extract hidden relationships and to discover novel knowledge, i.e., identifying key genes in biological pathways involved in drug response and adverse reactions. For this aim, we built a biological multilayer network comprising genes, drugs, diseases, and their associations extracted from a public database. Then, we analysed the multilayer network by applying a community detection algorithm, enabling the identification of essential genes from gene–disease–drug communities. After that, we used the identified list of genes from the communities to perform pathway enrichment analysis (PEA) to figure out the biological function affected by the selected genes. In particular, the identified genes are detached from their biological context, making it impossible to know in which biological mechanisms and functions they are involved. To understand which biological mechanisms are affected by these communities of essential genes, it is mandatory to link each gene to the opportune biological reference context by means of a pathway enrichment analysis (PEA). PEA links genes and groups of genes to the influenced biological pathways responsible for disease development, adverse drug reactions, as well as the different overall survival rates of patients treated with the same drugs. Thus, the new knowledge allows the development of new treatments that are more effective than drug repositioning strategies, in addition to realizing more adequate drugs for reducing or, even better, eliminating the onset of possible adverse drug reactions.

## 2. Background on Multilayer Networks

Multilayer networks have emerged as a powerful framework for modelling and analysing complex systems with multiple types of interactions. Unlike classical networks, which only consider one type of relationship between nodes, multilayer networks present the interdependencies between the entities of a system and the interacting layers [17]; see Figure 1 for a complete example.

Formally, a multilayer network can be traced back to a set of nodes, edges, and layers that take into account the physical and functional relationships between them [16,18].

In particular, each layer in a multilayer network represents a specific aspect or type of relationship between nodes. For example, in a social network, different layers can represent friendships, professional connections, or family relationships. Each layer can have its own set of nodes and edges, and there can be connections between nodes across different layers.

These networks provide a realistic representation of complex real-world systems, finding their way into practical applications in various domains, such as social network analysis, transport organization, biological systems, and technological networks [15,19]. Formally, a multilayer network graph may be described as a tuple $G_{ml}$ = $V_{L},E_{intra_{L}},E_{inter_{L}xL}$, where $G_{ml}$ is multilayer graph, $V_{L},E_{intra_{L}}$ is a set of nodes belonging to each layer, $E_{inter_{L}xL}$ is a set of edges belonging to each layer, and L={0,1,...l} is a set of layers. For each layer *k*, we have a graph $V_{k},E_{intra_{k}}$ (intralayer edges), and for each pair of layers, *k*, *h*, we have a set of edges (interlayer edges) $Einter_{v}xk$ connecting nodes of the layers v and k [20].

Examples of multilayer networks come from many different fields, from social network analysis to biological networks. For instance, Figure 1 represents an example of a biological multilayer network representing the interplay among diseases, genes, and drugs. Multilayer networks provide a powerful tool for analysing complex genetic and clinical data in pharmacogenomics, enabling better predictions of drug response, identification of drug targets, and acceleration of drug discovery and development processes. Multilayer networks can be used for general data analysis in pharmacogenomics. They can integrate and analyse large-scale genomic, transcriptomic, and proteomic data to uncover hidden relationships between genes, proteins, and drug responses. This can lead to a better understanding of the underlying mechanisms of drug response and aid in the development of personalized treatment strategies.

Furthermore, multilayer networks can be used to predict how individuals will respond to specific drugs based on their genetic information. By training the network on a dataset of patients’ genetic profiles and corresponding drug responses, it can learn complex patterns and make predictions for new patients. This can help identify individuals who are likely to experience adverse drug reactions or those who are more likely to respond positively to a particular medication. Multilayer networks can help identify genetic markers associated with the risk of adverse drug reactions (ADRs). By integrating genetic and clinical data, the network can identify patterns that link specific genetic variations to ADRs. This information can be used to develop personalized medicine approaches, where patients at higher risk of ADRs can be identified and alternative treatment options can be explored. Multilayer networks can aid in the identification of potential drug targets by analysing genetic data and can assist in drug discovery and repurposing efforts by analysing genetic data and identifying potential drug candidates. By integrating information from various sources such as gene expression, protein–protein interactions, and biological pathways, the network can identify key genes or proteins that play a crucial role in disease development or drug response.

## 3. Material and Methods

In order to apply multilayer network formalism and pathway enrichment analysis, with the goal to improve knowledge in the pharmacogenomics field, we design a methodology that comprises four steps:The building of a biological multilayer network comprising genes, drugs, diseases, and their associations extracted from the BioSNAP database;The analysis of the multilayer network by applying a community detection algorithm;The identification of essential genes from gene–disease–drug communities;Performing pathway enrichment analysis (PEA) to figure out the biological function affected by the selected genes.

Figure 2 summarize all steps.

### 3.1. Case Study

We considered the following datasets from the Stanford Biomedical Network Dataset Collection (BioSNAP) [21]:*Drug–Drug Interaction (DrDrI)* network of interactions between drugs, approved by the U.S. Food and Drug Administration (FDA): 1514 nodes and 48,514 edges.*Disease–Disease (DD)* network of interactions between 6878 inherited nodes and 6877 inherited edges.*Gene–Gene (GG)* network of interactions between in 25,825 inherited nodes and 208,836,746 inherited edges. The nodes are given by NCBI Entrez Gene IDs.*Disease–Drug Association (DDrA)* network, a set of curated relationships between diseases and drugs: 5535 disease nodes, 1662 drug nodes, and 466,656 edges. The diseases are given by DOIDs, i.e., Disease Ontology terms.*Gene–Disease (GDA) Association* network, a set of relationships between genes and disease: 7294 gene nodes, 519 disease nodes, and 21,357 edges.*Gene–Drug Interaction (GDrI)* network, a set of relationships between genes and drugs: 3648 gene nodes, 284 drug nodes, and 18,690 edges.

We build a multilayer network with three layers obtained from the DDI, DDr, and GG databases. Then, we add interlayer edges by considering the DDrA, GDA, and GDrI databases. Finally, the resulting multilayer network, that, for convenience, we called the GDD multilayer network, consisted of 52,640 nodes and 208,892,137 interactions, of which 506,703 *inter*edges exist. At first, we performed a topological analysis on the GDD multilayer network. The network analysis was performed using the multinet R package [22] (for complete details on multilayer network analysis, see [22]). Table 1 summarizes topological measures computed using GDD on the multilayer network for each layer. Table 2 summarizes topological measures computed using GDD on the multilayer network for layer comparison. The first part of the value indicates the type of comparison function (Jaccard, Coverage, Simple Matching, Russell Rao, Kulczynski, Hamann), and the second part indicates the configurations to which the comparison function is applied.

Table 3 summarizes the distribution dissimilarity computed using GDD on the multilayer network (notice that these are dissimilarity functions: 0 means the highest similarity) Table 4 summarizes the statistical degree correlations computed using GDD on the multilayer network.

### 3.2. Community Detection on GDD Multilayer Network

Once built, we analyse the GDD multilayer network by applying one of most useful exploratory technique for network analysis, i.e., community detection. Community detection is considered a first step in understanding network analysis and community structures, defined as groups of nodes that are more densely connected than the rest of the network, and represent significant characteristics for understanding the functionalities and organizations of complex systems modelled as networks. Thus, community extraction provides the identification of densely connected nodes within multilayer networks that play significant roles in the structure–function relationship. For this study, we selected Infomap [23] because, according to the literature, it outperforms other community detection methods for multilayer networks [24].

Then, we applied Infomap on the GDD multilayer network, obtaining 153 communities. Infomap extracted three typologies: (i) communities containing genes, diseases, and drugs; (ii) communities containing diseases and drugs; and (iii) communities containing genes. For our aims, we focus on the first typology of communities containing genes, diseases, and drugs. Then, we selected the top 10 communities, i.e., the communities comprising interlayer relations, for example, gene–drug and gene–disease relations. In Table 5, we reported the list of genes belonging to the top 10 communities.

### 3.3. Pathway Enrichment Analysis

Pathway enrichment analysis (PEA) helps researchers comprehend the biological meaning of gene lists obtained from high-throughput experiments, such as RNA sequencing, genome-wide association studies, or proteomics. These experiments identify genes, including proteins and metabolites, that differ between the conditions of interest. However, this gene list alone is insufficient to understand the biological differences between these conditions. Therefore, PEA assists researchers in interpreting large gene lists and developing hypotheses about the underlying biology [25].

To identify the biological mechanisms and/or functions affected by the identified communities of genes, we used the Reactome pathway database [26]. In particular, we describe the enrichment performed using the communities with identifier 10 using the software tool BiopaxParser (BiP-v.1) [27].

Table 6 reports the enriched pathways using the list of proteins belonging to community 10.

Next, we used BiP to know which pathways are influenced for each input gene. Table 7 presents the relation between genes and affected pathways. Inside community 10, a total of 36 genes are a member of layer 5, e.g., the disease–gene layer, and layer 6, e.g., the drug–gene layer, revealing the multiple roles of a gene through different pathways. Analysing Table 7, it is worth noting that the activity of the *metabolism of proteins* pathway, a well-known pathway related to adverse or normal drug responses, as well as to disease progression or decline, is regulated by the interactions of more multilayer genes, namely *P43088, P15170, P18509, P05546, Q9Y277, Q02817, Q13285, O75976*, and *P15328*, reinforcing the benefits of the multilayer formalism to represent complex networks.

### 3.4. Results and Discussion

Pharmacogenomics is a complex field where the drug response of the living organism is due to the interactions of several different biological entities like genes, enzymes, and small and large molecules that cooperate in a synchronized fashion to accomplish the task. Multilayer network representation allows for more comprehensive and realistic modelling of these heterogeneous interactions than traditional ones [28]. In addition, multilayer networks enable the identification of multilayer communities that are a bunch of genes more densely connected among them and the correlations through the different layers: information that can be used to perform PEA to comprehend the affected underlying biological mechanisms.

Performing PEA using the detected gene communities from layers 10 enriches several biological pathways, as reported in Table 6. Analysing the content of Table 6, it is worth noting that the enriched pathways present multiple intertwinements among them, some of which are more explicit than others. We conducted a literature search to explore the possible connections between the results of the protein enrichment analysis. According to Bhardwaj et al. [29], leishmania alters various signalling pathways to survive, which is in line with the other enriched pathways 6,7,8,9. Additionally, Kaiser [30] found that cyclic nucleotides such as cAMP and cGMP are crucial for parasitic proliferation and regulate functions such as auditory and olfactory senses [31]. Also, Rho GTPases play a role in host–pathogen interaction by controlling innate and adaptive immune responses. Pathway 1 in Table 6 is another signalling pathway that leishmania affects, as described in [31]. Moreover, Schlessinger et al. [32] explain the vital role of the mediator of Rho GTPases in the WNT signalling pathway. Finally, Kikuchi et al. [33] describe the regulation of WNT signalling pathways through post/translation modifications, while Li et al. [34] provide details on the role of RUNX1 in promoting tumour metastasis by activating WNT.

Table 7 clearly displays the association between genes and pathways, emphasizing that a single gene can be involved in multiple pathways. The enrichment was calculated by implicitly incorporating topological and structural network properties, resulting in improved enrichment outcomes, as opposed to using more general genes, as described in [27].

In Figure 3, community 1 is represented as a network. The green nodes with red labels in the network correspond to the genes listed in Table 7. These genes play a crucial role in the network’s connectivity and are known as **hub genes** in the literature. For instance, if we remove *O95182*, *P15328*, and *P09012*, the network loses its complete connectivity. To learn more about the role of the three hub proteins, we searched the Reactome database. We used the Reactome web pathway browser and found out that the three hub proteins are a part of the metabolism pathway. Specifically, all three proteins have an impact on the citric acid (TCA) cycle and respiratory electron transport pathway. Moreover, proteins P09012 and O95182 regulate respiratory electron transport, ATP synthesis via chemiosmotic coupling, and heat production by uncoupling proteins pathway. This highlights the importance of multilayer modelling and enables the selection of more relevant genes from the network for performing PEA.

In addition, Table 7 includes some genes that are not part of community 1, featured in Figure 4.

If we rely solely on a traditional network representation, we may overlook crucial information, such as the fact that gene *P18509* does not belong to community 1. However, through a multilayer representation, we can observe that gene *P18509* interacts with *P10109*, as illustrated in Figure 3.

The reason for this is that both genes affect the same category of pathway related to the metabolism. In conclusion, the use of multilayer networks to represent interactions among heterogeneous data is a novel approach, especially in the field of omics. A multilayer approach can help researchers capture more information and obtain a more accurate understanding of gene interactions. In the literature, Shang et al. in [28], propose a multilayer network representation learning method for predicting drug–target interactions. This method integrates information from different networks, reduces onise, and learns the feature vectors of drugs and targets, overcoming the challenges of integrating multiple data types and managing network noise. Using a multilayer network to infer new relationships among genes, diseases, and drugs is at its early stage and is a continuously developing field. This limits the possibility of validating the proposed method by comparing it with existing methodologies. Using the proposed method, we discovered potential new relationships between leishmania and different signalling pathways: results possible only through multilayer representation. This could help researchers to identify drugs targeting specific biological functions affected by the enriched pathways. Investigating leishmania is particularly important in the context of travel medicine. Berman reviewed several aspects of diagnosis and treatment for leishmania in [35]. With our method, we could determine which drugs could contrast the damage caused by leishmania infection.

## 4. Conclusions

In this work, we explore the application of network analysis in the pharmacogenomics field. In particular, we used multilayer network representation to model the interaction among genes, drugs, diseases, and their associations. Then, we analysed the network by applying a community detection algorithm to discover the top communities. Finally, we used the identified list of genes from the communities to perform pathway enrichment analysis (PEA) to figure out the biological function affected by the selected genes. The results demonstrate that the genes forming the communities extracted from the multilayer network regulate the activity of the *protein metabolism* pathway related to adverse or normal drug response, as well as the progression or decline of disease, demonstrating the advantages of multilayer formalism to represent pharmacogenomic domains.

## Figures and Tables

**Figure 1 genes-14-01915-f001:**
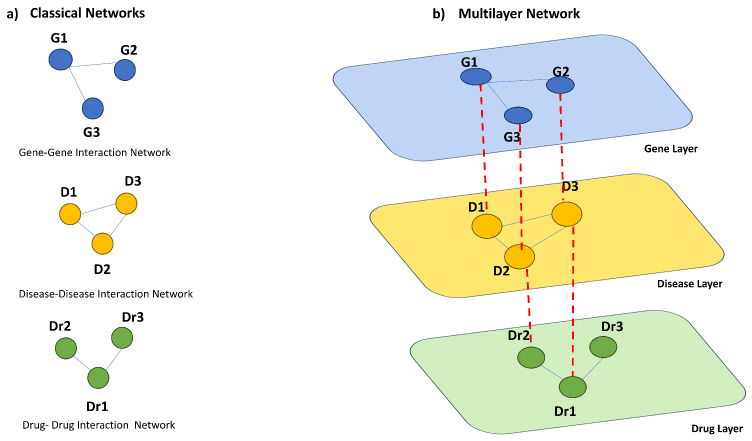
Examples of classical and multilayer networks. The figure shows two toy examples of a classical biological network (**a**) and a multilayer network (**b**). The first example (**a**) reports three different networks, i.e., a gene–gene interaction network, disease–disease interaction network, and drug–drug interaction network. In the second example, each of those networks represent a distinct layer in the multilayer network. The nodes of the multilayer network are the genes, the diseases, and the drugs, all discriminated by belonging to the respective layer. The *intra*-edges represent the gene–gene, drug–drug, and disease–disease associations, while the *inter*-edges are the gene–disease, gene–drug, and disease–drug associations.

**Figure 2 genes-14-01915-f002:**
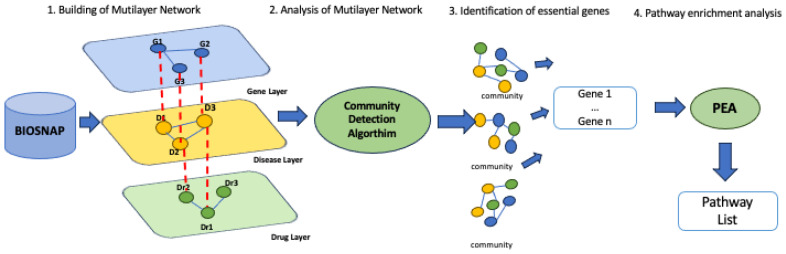
Methodology workflow.

**Figure 3 genes-14-01915-f003:**
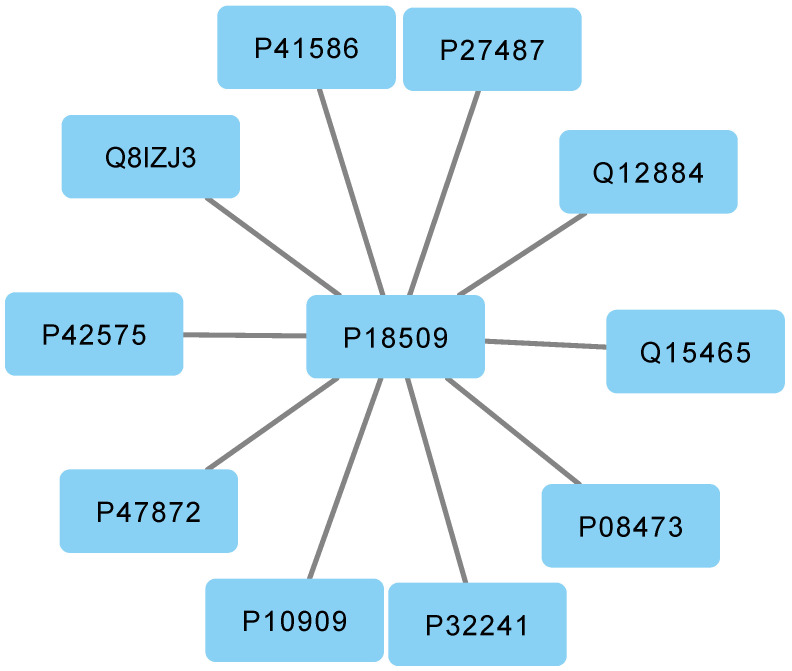
Interactions among genes in community 2. The figure displays a network diagram depicting the interactions among the genes that belong to community 2.

**Figure 4 genes-14-01915-f004:**
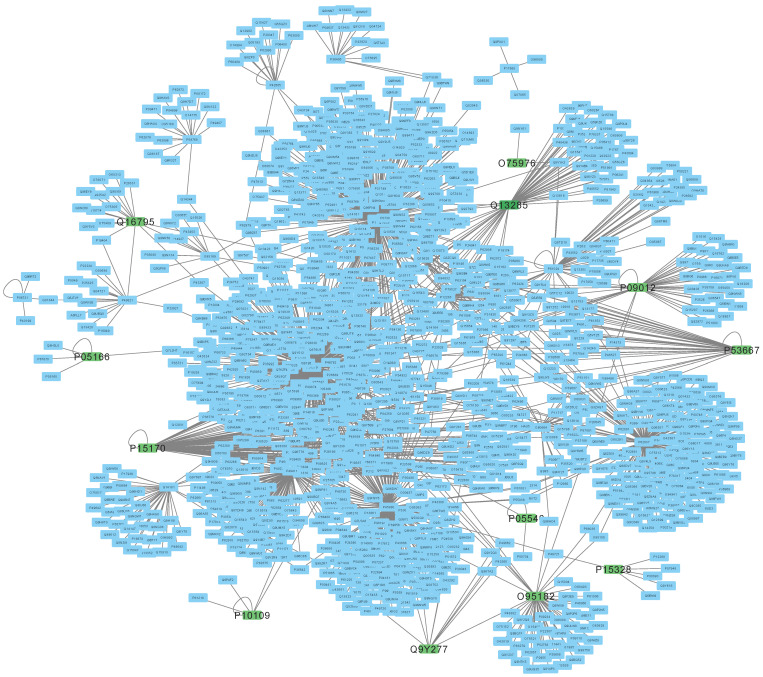
Interaction among genes in community 1. The figure displays a network diagram depicting the interactions among the genes that belong to community 1. In the network, green nodes with red labels indicate the genes listed in Table 7.

**Table 1 genes-14-01915-t001:** Topological measures computed using GDD on the multilayer network for each layer.

Network Measure	Layer 1	Layer 2	Layer 3
similarity between layer summaries	1.126	4.936	4.888644
layer min degree	1	1	1
layer max degree	89	443	696
layer mean degree	1.999	64.087	68.301
layer sd degree	3.239	68.339	90.953
layer skewness degree	11.373	1.5809	2.187
layer kurtosis degree	221.854	5.724	8.888
layer entropy.degree	1.126	4.936	4.888
layer CV degree	1.619	1.066	1.331
layer jarque bera degree	13,874	1098.965	4780.306

**Table 2 genes-14-01915-t002:** Topological measures computed using GDD on the multilayer network for layer comparison.

Network Measure	Value
Jaccard actors	0.268
Jaccard edges	0.2
Jaccard triangles	0.01
coverage actors	0.34
coverage edges	0.2
coverage triangle	0.1
sm actors	0.658
sm edges	0.999
sm triangles	1
rr actors	0.061
rr edges	8.523 exp −5
rr triangles	3.064 exp −0.9
Kulczynski2 actors	0.3374
Kulczynski2 edges	0.2
Kulczynski2 triangles	0.1
Hamann actors	0.317
Hamann edges	0.998
Hamann triangles	0.999

**Table 3 genes-14-01915-t003:** Distribution dissimilarity computed using GDD on the multilayer network.

Network Measure	Value
dissimilarity degree	0.232
KL degree	0.57
Jeffrey degree	0.949

**Table 4 genes-14-01915-t004:** Statistical degree correlation computed using GDD on the multilayer network.

Network Measure	Value
Pearson degree	0.153
rho degree	0.220

**Table 5 genes-14-01915-t005:** Top ten communities.

Community	Genes
1	*P48169 Q15822 P08172 P08908 P25100 P28222 P19320 Q01959 O15399 Q9NYX4 P48051 P32297 P23560 P17787 P01189 P47870 P31645 P08173 O00264 P30939 P54219 P41145 Q9H3N8 P28223 P07550 P18825 P41143 Q12879 P28566 O60391 O14764 Q05586 P13945 P28476 P11229 P42263 O14732 P28472 P08912 O60858 Q8N9I0 Q13002 P30926 P28221 P08913 P21728 P34903 P14867 P48058 Q9UK17 Q16445 P23416 P46098 P17405 Q09470 P42262 P02708 P41595 P23415 P18507 P18505 P30049 P43681 O75311 Q8TCU5 Q9UGM1 Q96RJ0 Q9UHC3 P50406 Q15825 P02763 P28335 P19652 Q13224 P31644 P48549 Q99928 Q9NPC2 A5X5Y0 Q9GZZ6 P42261 P35368 P03886 P48167 P24046 P41146 Q9Y2I1 Q05901 Q99720 A8MPY1 P30536 P21918 Q9UN88 P35367 P20309 Q05940 P14416 P08588 O95264 P78334 O14649 P34969 Q9NZV8 Q8N1C3 P47898 P35348 P18089 P35372 P35462 Q14957 P23975 P47869 P36544 P23763 P21917 P30988 O00591 Q8WXA8 P20711 P98194 Q9Y5N1 Q70Z44 P13498 P30532 O94956 Q14542 P63027l*
2	*P09001 Q8N6M6 Q96P70 Q9NZV1 H0YKE7 Q9UIX4 O95267 O95155 E9PMR6 Q9C0B9 O14950 Q96KV7 Q4LE39 O95704 O60610 Q9H2X9 Q14376 Q14376 Q5T7N3 Q96AP7 Q01726 Q01726 Q6ZS17 Q14554 P04053 P04053 Q9NQW5 Q9BSH5 Q13641 O95822 Q9UK23 Q9HAV5 D018197 D009459 Q9BPW8 A0A087WYT5 F5GX24 Q96P09 Q7Z5S9 Q96NT5 Q0VDG4 Q9C0G0 P47897 O95236 P51805 Q5VZE5 Q13535 P43489 B7Z503 O95229 O96013 Q8N1G4 Q07065 Q13467 Q7Z7J9 P34130 Q9UBV2 B4DK25 Q8TCA0 Q9H081 Q53H47 P43250 Q9UKK6 B5MCN0 Q9NWW6 O75746 O75746 Q9NQZ2 Q8TC12 Q8TC12 P35080 O00267 Q9C019 P50749 P14384 Q8WUA2 P84243 Q86SG2 A0A0A0MSX3 Q96C11 Q14181 Q14181 Q9P2C4 O15360 Q9H116 Q9H7F0 Q9NZU0 Q8WWL2 Q8WWL2 Q15034 Q96BZ9 A0A0C3SFZ7 Q8IVM8 Q96FL8 Q96FL8 Q92804 J3KNJ3 Q96CU9 P25490 Q15561 Q15543 A0A0D9SEJ5 Q06546 Q16774 Q14CZ0 P61088 Q96GD3 Q15434 Q9Y5G5 Q8N6T3 P35498 P35226 P10997 Q5VWQ8 A0A0C4DGU8 E9PAV3 Q96HZ4 P83111 Q5JTH9 E7EUK8 O15229 O95047 Q13439 P63162 O75431 A0A0A0MTN6 P50222 Q8NBX0 Q15399 P22362 O43312 Q9BWX1 P48431 Q15831 Q9BWM7 Q8IX12 Q01844 Q92611 Q9NYU1 P51797 Q53GI3 P43088 P43088 Q16775 Q16775 Q9BZF9 P57723 Q8TC71 Q8WUM4 Q9BV90 A0A0G2JNH5 Q9BRJ7 Q8TEZ7 Q6P4H8 Q5VWK5 Q9Y3A4 P49366 P49366 Q9HC10 O60575 P78330 P78330 O43766 Q7L7X3 Q5TB30 P20962 P78410 O76024 Q13625 Q13469 Q96GM1 Q9H582 Q8TCX1 Q9HC96 P35626 P35626 Q9H089 B7Z4U2 Q13427 Q13427 Q5BKX8 O15164 P17568 P17568 Q71UI9 B7Z673 O15119 Q76EJ3 Q5XKR4 Q9Y2L1 A0A0A0MSZ8 Q7Z699 O14713 P53675 Q9P255 Q9P212 P29372 O95263 O95399 Q96PC3 Q9GZR7 Q9Y4Z0 A0A0A6YYA2 Q6UXH1 Q9NV66 Q969Z4 P29084 Q96IF1 O75425 DB08912 DOID:9811 DOID:9822 DOID:9842 DOID:9820 DB06271 DB08827 DB09033 DB00089 Q9BZQ8 DB08819 DB08819 Q86WU2 Q8WXK3 D001749 Q86VW0 D006073 Q96PV7 D005076 Q9P107 Q8NEB5 D057049 C535575 Q02410 D066126 D009669 P26196 D009784 Q8TF05 D001988 D052439 D006463 Q9NRX2 P36639 D001172 Q9UL25 D015866 D007410 D014062 D000707 Q6P1J9 J3KSP0 D000138 Q9BYG3 Q15907 D003248 P09543 D013281 A0A087WWW8 E9PN67 Q9UBQ7 D018777 Q9UKG1 D003093 Q9Y2P5 P23297 P23297 A0A087X1W2 Q64LD2 D007822 D000267 D002761 D002769 Q13324 Q5HYK3 A9UHW6 D006457 D002114 D012769 D013086 D020767 D003327 D015267 P22557 D014657 D054038 P55042 D008108 D019337 D002105 Q15691 D019575 Q9H4T2 D057772 D003324 Q06033 D029424 D003928 D014555 D010392 Q15014 Q8WXH5 D002292 D015464 A0A088AWP5 Q9NZ08 Q14739 I3L419 D011529 Q8N6M3 Q8WYQ9 Q16762 Q16762 D000402 Q71F56 Q8IYT3 D009468 Q7Z4F1 P80162 Q9BTT0 D005483 P05496 Q9Y6J6 D052016 Q96GQ7 P07858 C536522 B0QYL9 D010610 D009447 D020773 Q9H3U5 Q15836 Q9BXX0 D012170 B7ZAA0 B4DRX9 D014570 D000419 P09417 P09417 D003424 D005119 Q3B7J2 P09661 D006978 Q99832 D017119 Q5TCZ1 D019584 P12829 Q06141 Q9H361 O43924 Q9HCK4 D020246 D007511 D006504 D000749 Q01850 D015529 Q13206 Q68D51 Q9UJ72 Q16864 D010003 D019214 Q15648 D006996 Q56NI9 D007565 D001649 Q53S58 D005706 Q4LEZ3 P36382 D004827 D005910 O15118 D006261 D020250 P04049 D007638 O43739 O43739 Q7L273 D003693 Q9C0H6 Q8NAB2 Q96GK7 E9PKC0 D001766 Q08211 P36021 P82987 D006985 P49902 P49902 Q15393 J3KRL0 D006099 Q07075 Q07075 A0A087WXE9 Q9H6X4 D013959 Q9NPJ1 Q8NG11 D055665 Q96JQ0 D004681 P30050 O00167 Q9XRX5 Q8TDY8 Q9BY21 C535841 Q5TAQ9 D020896 Q2V2M9 D001660 D015427 Q8WY36 Q5VXI4 Q9BS34 O76080 D008654 D003645 D013276 Q16778 Q9HAW4 D009901 P04350 D017544 C538090 D007239 A0A087X004 P15170 P15170 Q9Y4B6 D015658 Q9NY84 P41250 D033461 Q9H857 Q9BUR5 Q9UKX2 A0A087WUN7 J3KNS1 D013272 Q9UPY5 Q8N392 Q9H8E8 E9PLK3 D001528 Q13275 Q7Z7E8 D009402 Q96FG2 O95996 Q9Y2B9 Q9BUA6 Q99687 Q9UHE8 D011695 Q96DC8 Q9ULC0 Q9H1K4 Q9H1K4 D008661 Q8N3C0 D001424 Q9H694 C9J975 Q9H2S9 Q6P444 D019694 Q8TE82 Q7Z5R6 Q8TDN4 D015210 Q9UM01 D017379 O15230 D054537 Q9H2K0 F5H4S8 Q9Y2L6 Q96PB1 Q13595 Q15406 Q8NDL9 D000544 Q14919 A0A0A0MRS4 D020261 P02649 Q9HB90 P46060 O43365 Q92888 Q12815 D020275 D014571 Q00056 D013923 Q16795 Q16795 D003876 P49682 Q16513 Q9ULG6 Q9ULC3 Q969S3 E9PCD7 146850 Q96A59 Q9P016 Q96JB5 Q8IXT1 Q6P5X7 Q9NY12 Q6W4X9 O75508 Q14CZ7 O95425 Q7L2H7 Q3KQU3 Q92781 Q92781 D009364 Q9BZL3 A0A0A0MSU4 Q9NXL6 Q8TE73 O43293 O00421 O00624 O95490 Q6PI25 D012130 A0A087WW65 Q8NG68 P08700 D003872 P15812 D054198 A8MQT2 D013964 Q9BV36 Q6ZRI6 Q9NV12 D014985 Q8NDI1 D018771 P61247 Q8IZJ1 D6RF85 Q9BWE0 Q9P2K6 Q6R327 Q8N5H3 Q8TB22 Q8NB14 Q9HA82 Q6ZVK8 O95613 E9PBC5 P28845 Q8TAP4 Q96A57 Q9BVJ7 D016640 Q6VN20 Q6DN14 Q969Q4 Q9Y2B0 P06746 O60704 D011225 Q9Y2G3 Q9H8P0 Q9H8P0 P21452 G3V2U7 Q96S44 P16144 P61225 Q99595 Q92503 A6NGC4 Q8NFT2 D003929 Q9H981 D056784 Q9NRZ5 O43280 Q96PD2 Q8TAD4 Q9BY49 D012185 D006258 Q96Q15 P02549 Q8IY21 P23760 A0A087X2I7 Q6NYC1 Q92902 E9PCV4 Q99801 D002561 D009220 Q9Y232 B4DFF3 Q13177 D016849 Q99969 Q5JTW2 D004417 E9PHE4 D053099 A0A087X2B0 Q9P2Y5 Q96T68 D014456 E9PH60 Q9BTT4 A0A087X1U6 Q76M96 A0A0C4DG21 Q96D31 Q2KHN1 P63244 A0A0A0MQS7 Q6P5W5 Q9NPG2 Q9BVC3 P48634 Q14674 Q9ULX6 Q8TB24 D008545 D015878 F8W7W8 P48742 P18509 P18509 Q96FW1 Q06495 Q6YN16 D010148 Q15020 Q8TEB1 O00161 O14521 D010018 D010024 Q96PZ0 D011833 Q92890 D009008 Q16659 P24666 P24666 Q13488 G3V2R1 Q96AW1 Q8NFY4 Q9NP97 D009404 D003456 E7EVZ1 J3QQY4 P34059 P34059 Q02543 E9PPM8 Q9Y5K6 Q9H2U2 D014605 P05546 P05546 B7Z1U7 D009846 Q8N6M8 D018269 P06132 P06132 P35232 D002637 Q9NRR2 Q16842 Q5TC84 Q9NWU1 D008546 Q8N1N5 D006932 Q10981 P55735 Q9NPI1 D001168 Q8N4V1 Q6ZMW2 Q96NY8 P51178 Q92622 Q9UBI9 P21796 D005923 F5H5D3 Q99623 Q9HBJ8 O00299 Q6V0L0 P55036 Q9Y3C7 Q9Y2I2 Q9Y376 Q99715 D005921 Q8N6G5 B0QY90 D008664 Q9UK45 Q96AG3 Q9NRM7 D006983 P32302 A0A0G2JP62 P23497 Q9BRX5 Q86WV8 O75342 D012164 D008223 D004412 Q9NR00 D053627 D005157 D011115 Q02446 P82932 B3KVL5 Q14563 D006471 Q96QF0 Q969M7 Q5SSJ5 P55795 D020434 P60983 P30038 P30038 Q9BRF8 Q6IQ32 Q9NVJ2 P02774 Q3MIP1 P14317 Q8TC20 P04628 D020138 D007681 Q8TE02 D006977 P84090 Q8N6C5 D6R938 O75386 Q9H9H4 Q5VY80 O95461 Q8WVC6 Q6ZQR2 Q96DX8 D006940 Q8WXK4 P49441 Q9UBV7 O94855 Q86VR8 A0A087X0D9 Q01970 Q9Y5P3 P54802 P54802 D016403 Q5SNT2 Q86SF2 O94810 P29400 O43286 O00244 O00244 Q9BXB1 Q9H841 G3V1T9 Q9P0M6 Q12972 D042822 Q9ULV1 P98168 P41231 P41231 P15924 Q13045 D016889 P20290 Q9BY84 D005158 P56749 Q8TDM6 P48067 P62701 Q86WK7 F5H8H2 D009358 Q9H160 Q86Y78 C9JYQ9 O43660 Q9NR61 D011697 D009128 P09012 P09012 Q9GZX9 P35908 Q9Y6X0 P48059 Q9H9C1 A0A0A0MRE5 Q96HR9 Q5QPE7 Q8ND07 D016770 Q9NZZ3 Q9H999 Q01484 D005891 Q9GZN2 A6NHQ4 Q9H6D3 Q99708 P10253 D004646 P0CG34 P26599 Q9Y2S0 P63261 Q16719 D055009 Q9NZJ4 Q5U651 G1UD79 Q9BSK4 Q9UGM6 Q9NQZ7 Q9UGP5 Q9BS26 D007642 P09668 D054218 D016553 P14920 Q96GW7 A0A0A0MS77 O14638 Q9UH17 P78317 Q9NVK5 F5H3L1 Q9UI17 A0A0C4DGP2 Q9Y4E8 Q8TD43 Q8N3V7 P51800 P51800 D050500 Q99543 O60244 Q63ZY3 Q9Y6C9 P35268 Q9Y277 Q9Y277 Q9H6S0 D015473 P20916 Q9Y566 Q96LJ7 Q93070 D006947 O95571 O75146 A0A0A6YYA7 D016534 Q8N436 Q99675 P02747 Q8IUC4 A0A0A0MR67 P51606 P51606 Q9HAW0 P22674 P58876 O15273 D012851 Q13772 E9PCH5 Q96AQ6 Q70IA6 Q99417*
3	*DB01611 P17900 O60603 O75116 P01308 O00206 P17612 Q9NR96 P61925 Q9NYK1 P31749 Q04771 P43080 Q13464 Q9Y6Y9 P31751 P62942 P49841 Q08209*
4	*P14555 P02788 P04054 Q16706 P63208 Q9UKM7 DB03414 Q9NZK7*
5	*DB03880 Q9ULZ9 P09238 DB01197 Q9UKQ2 P39900 P08254 P09237 P14780 P03956 P51512 P08253 Q8N119 Q9UNA0 O60882 O60882 Q9Y5R2 Q9H306 P51511 P24347 Q9NPA2 P50281 P45452 P22894*
6	*P01133 P04818 Q8NBP7 P05106 P54760 Q7LG56 Q99808 Q99062 P27707 P33151 P05091 P09871 O00142 P12318 P15309 Q16552 P32321 P06746 P12821 P30085 P26358 P62993 P00797 P31785 P14784 P02747 Q9NRF9 P56282 Q07864 P16220 P16066 P31994 Q05932 P08514 P00533 P36952 P00736 P24385 P19971 P04234 P32320 P12314 P12314 P02746 P20594 Q9UNI1 P01031 P05186 P78559 P01589 P15391 P31350 P08473 P24158 Q03393 P11836 P20701 O60493 Q9BYF1 P08246 Q16854 P23919 P22413 P19235 Q15303 P04183 O14788 P22102 P02745 Q92820 P31995 Q9NNW7 P31939 P08637 P0CG22 P04626 O75015 P00813 Q9H252 P23921 O00764 P00374 P04229 P0C0L5 P09884 P17342*
7	*P22303 P21964 Q92952 DB06218 P06276 Q86W47 Q9UQD0 P21397 P27338 Q16558 Q9NY46 P07686 Q9Y5Y9 Q9H2S1 Q9NPA1*
8	*TP61457 Q9Y619 P05089 P29475 P78540 Q8WY07 P50440 Q9BXI2 P00439 P13716 P52569 Q15046 O95190 P35228 P01270 Q96A70 P00480 P04424 O43246 P30825 P29474 P54368 Q9UMX2 P20823 P35222*
9	*P35916 P20648 P49286 P46059 P10636 P16234 P06133 P09172 P11473 P33260 P16444 Q8TCC7 P02768 O15554 Q9Y6L6 Q9UM07 P24530 Q15858 Q9HCR9 Q9BQB6 P01023 P25021 P11712 P04629 P10635 O15111 P13569 P33402 Q9NPD5 Q16696 P31513 Q9Y5Y4 P54855 Q12809 P15538 Q6VVX0 Q9NSA0 P31639 P48039 Q9H244 P50225 P43116 Q15166 Q13956 Q969P6 Q92769 P05181 P08684 Q99250 P22309 Q4U2R8 P33261 Q9HB55 P09619 Q16853 P25024 Q9UI33 P19835 Q02928 P35499 P10632 P23219 O76074 P43115 DB00192 Q14524 Q16850 Q92753 P16662 P18440 P30711 P30556 Q9UQQ2 P18545 Q9UHC9 O43923 Q14123 P15382 P08174 Q08345 P04156 P09086 P15502*
10	*A8TX70 A9YTQ3 A9Z1W8 A9Z1W9 B0FP48 B0I1T2 B0QY83 B0QYH2 B0QYP2 B0YIZ1 B1AKI9 B1AKP1 B1ALC3 B1ALM3 B1ALM5 B1AN62 B1AVT0 B1B0D4 B1PBA3 B2RUZ4 B2RXF0 B2RXF5 B2RXH4 B2RXH8 B3KP42 B3KQ25 B3KRF7 B3KT41 B3KUS5 B4DDH2 B4DED4 B4DFN8 B4DGD8 B4DGE2 B4DGY6 B4DH95 B4DJV5 B4DK72 B4DL54 B4DLN5 B4DLR2 B4DMT0 B4DP31 B4DPR9 B4DRN8 B4DRP8 B4DS77 B4DSF2 B4DT06 B4DWF2 B4DWR3 B4DX60 B4DXD0 B4DXG3 B4DY09 B4DY77 B4DZ96 B4DZG1 B4DZG7 B4E059 B4E0T2 B4E128 B4E2Q0 B4E3E2 B5M0C0 B5MBW9 B5MBY4 B5MCQ6 B5MCR8 B5MCV2 B5MCW3 B5ME49 B5ME97 B7WNR7 B7WPH3 B7WPL0 B7XGB9 B7Z1M9 B7Z1N8 B7Z2R7 B7Z2U2 B7Z3H4 B7Z3L0 B7Z3L6 B7Z4K4 B7Z779 B7Z7B0 B7Z7D2 B7Z7F3 B7Z8L1 B7Z964 B7Z9H7 B7ZAQ6 B7ZBD5 B7ZL88 B7ZL91 B7ZLH2 B7ZLJ8 B7ZLQ8 B7ZLU9 B7ZVY7 B8PRF2 B8ZZI5 B8ZZJ6 B9A047 B9EJG8 C3TX97 C535854 C535906 C535935 C536122 C536424 C536600 C536681 C536848 C536875 C537168 C537328 C537352 C537475 C537923 C538539 C562792 C563177 C563435 C563671 C564322 C564492 C564507 C566033 C566384 C566510 C566882 C567393 C567492 C567527 C567595 C567616 C567734 C567747 C567832 C9J167 C9J2Y9 C9J3P7 C9J3V5 C9J6L6 C9J7D0 C9J7S5 C9JBD0 C9JE40 C9JEH3 C9JER5 C9JFE4 C9JFV4 C9JFX4 C9JH25 C9JI98 C9JJX6 C9JLB7 C9JLW8 C9JME2 C9JN71 C9JNI2 C9JQI7 C9JQV3 C9JQY5 C9JR72 C9JRZ8 C9JXX5 C9JYJ0 C9K0J5 E0CX11 E1CKY7 E2JJS1 E2QRF5 E5RG02 E5RGB3 E5RGY0 E5RHT3 E5RIK9 E5RIX8 E5RJN2 E7EM93 E7EMB1 E7ENR6 E7EPC3 E7EPI0 E7EPP7 E7EQD3 E7EQG5 E7EQI7 E7ER45 Q9P0U1 Q9P0V8 Q9P0W0 Q9P126 Q9P127 Q9P1P5 Q9P1Q5 Q9P1U1 Q9P1W8 Q9P1Y5 Q9P209 Q9P215 Q9P218 Q9P242 Q9P253 Q9P258 Q9P265 Q9P286 Q9P289 Q9P296 Q9P298 Q9P299 Q9P2A4 Q9P2B4 Q9P2D1 Q9P2E3 Q9P2E7 Q9P2F5 Q9P2F6 Q9P2G4 Q9P2J2 Q9P2J9 Q9P2K3 Q9P2K5 Q9P2M1 Q9P2P1 Q9P2P6 Q9P2T0 Q9P2V4 Q9P2W3 Q9P2W6 Q9P2W7 Q9UBB6 Q9UBC5 Q9UBC9 Q9UBD0 Q9UBD3 Q9UBE0 Q9UBF1 Q9UBF2 Q9UBG3 Q9UBH0 Q9UBL6 Q9UBM4 Q9UBN1 Q9UBP0 Q9UBP6 Q9UBR2 Q9UBS9 Q9UBT3 Q9UBT6 Q9UBT6 Q9UBU3 Q9UBZ4 Q9UC07 Q9UD57 Q9UDW3 Q9UDY6 Q9UEG4 Q9UEW8 Q9UF02 Q9UF47 Q9UFB7 Q9UFC0 Q9UFD9 Q9UFE4 Q9UFN0 Q9UG01 Q9UGB7 Q9UGC6 Q9UGC7 Q9UGF5 Q9UGF6 Q9UGI0 Q9UGI9 Q9UGI9 Q9UGK8 Q9UGL1 Q9UGM6 Q9UGT4 Q9UGV2 Q9UH03 Q9UH36 Q9UH77 Q9UH90 Q9UH99 Q9UHA3 Q9UHA3 Q9UHA7 Q9UHB4 Q9UHB9 Q9UHC7 Q9UHE5 Q9UHF3 Q9UHF4 Q9UHF7 Q9UHI6 Q9UHJ3 Q9UHM6 Q9UHN1 Q9UHP6 Q9UHR5 Q9UHR6 Q9UHW5 Q9UI26 Q9UI30 Q9UI33 Q9UI38 Q9UI40 Q9UI47 Q9UIA0 Q9UID3 Q9UIG4 Q9UIJ5 Q9UIL1 Q9UIL4 Q9UIM3 Q9UIU6 Q9UIW0 Q9UJ14 Q9UJ71 Q9UJ78 Q9UJ90 Q9UJ96 Q9UJA2 Q9UJA3 Q9UJD0 Q9UJF2 Q9UJG1 Q9UJJ7 Q9UJK0 Q9UJL9 Q9UJN7 Q9UJQ4 Q9UJQ7 Q9UJT2 Q9UJW0 Q9UJW2 Q9UJX2 Q9UJX5 Q9UJX6 Q9UJY4 Q9UJY5 Q9UK05 Q9UK11 Q9UK28 Q9UK32 Q9UK59 Q9UK73 Q9UK96 Q9UKA1 Q9UKA2 Q9UKA8 Q9UKB5 Q9UKD1 Q9UKF2 Q9UKF5 Q9UKF6 Q9UKG4 Q9UKJ0 Q9UKJ5 Q9UKJ8 Q9UKK9 Q9UKM7 Q9UKN1 Q9UKN7 Q9UKR0 Q9UKR3 Q9UKR5 Q9UKR8 Q9UKS6 Q9UKT5 Q9UKT8 Q9UKT9 Q9UKU6 Q9UKU9 Q9UKX3 Q9UKX5 Q9UKZ4 Q9UKZ9 Q9UL16 Q9UL19 Q9UL42 Q9UL62 Q9UL68 Q9ULC4 Q9ULC6 Q9ULD8 Q9ULE0 Q9ULG1 Q9ULH4 Q9ULI0 Q9ULJ3 Q9ULK2 Q9ULK4 Q9ULK5 Q9ULL4 Q9ULL8 Q9ULM2 Q9ULM3 Q9ULM6 Q9ULS6 Q9ULT0 Q9ULU8 Q9ULV5 Q9ULW2 Q9ULW3 Q9ULX3 Q9ULX5 Q9ULZ0 Q9ULZ2 Q9UM07 Q9UM13 Q9UM73 Q9UM82 Q9UMF0 Q9UMN6 Q9UMQ3 Q9UMQ6 Q9UMR3 Q9UMR7 Q9UMS0 Q9UMX6 Q9UMY1 Q9UMY4 Q9UMZ2 Q9UN66 Q9UN71 Q9UN73 Q9UN74 Q9UN75 Q9UN88 Q9UNA3 Q9UNA4 Q9UND3 Q9UNE0 Q9UNG2 Q9UNI6 Q9UNN5 Q9UNQ2 Q9UNT1 Q9UNX3 Q9UNX3 Q9UNX9 Q9UNX9 Q9UNY4 Q9UPA5 Q9UPC5 Q9UPE1 Q9UPI3 Q9UPM6 Q9UPM8 Q9UPN9 Q9UPP1 Q9UPP2 Q9UPQ3 Q9UPR3 Q9UPS6 Q9UPS8 Q9UPT8 Q9UPT9 Q9UPV0 Q9UPW6 Q9UPX0 Q9UPX6 Q9UPZ3 Q9UQ05 Q9UQ10 Q9UQ13 Q9UQ49 Q9UQ72 Q9UQ74 Q9UQB9 Q9UQC2 Q9UQC9 Q9UQD0 Q9UQF0 Q9UQF2 Q9UQN3 Q9UQQ1 Q9Y210 Q9Y215 Q9Y217 Q9Y222 Q9Y224 Q9Y228 Q9Y231 Q9Y236 Q9Y238 Q9Y243 Q9Y253 Q9Y256 Q9Y259 Q9Y263 Q9Y264 Q9Y266 Q9Y267 Q9Y275 Q9Y284 Q9Y285 Q9Y286 Q9Y291 Q9Y296 Q9Y2A4 Q9Y2B2 Q9Y2B4 Q9Y2B5 Q9Y2D0 Q9Y2F5 Q9Y2F9 Q9Y2G1 Q9Y2G8 Q9Y2G9 Q9Y2H9 Q9Y2K3 Q9Y2K6 Q9Y2L5 Q9Y2L8 Q9Y2M0 Q9Y2M2 Q9Y2M5 Q9Y2P0 Q9Y2P4 Q9Y2Q1 Q9Y2Q9 Q9Y2T3 Q9Y2T6 Q9Y2T7 Q9Y2U2 Q9Y2V3 Q9Y2V7 Q9Y2W7 Q9Y2X0 Q9Y2X7 Q9Y2Y0 Q9Y2Y8 Q9Y2Z0 Q9Y303 Q9Y312 Q9Y328 Q9Y330 Q9Y333 Q9Y334 Q9Y336 Q9Y337 Q9Y345 Q9Y366 Q9Y394 Q9Y3A2 Q9Y3A5 Q9Y3B3 Q9Y3C5 Q9Y3C6 Q9Y3C8 Q9Y3D7 Q9Y3D8 Q9Y3D9 Q9Y3E0 Q9Y3F4 Q9Y3I0 Q9Y3L3 Q9Y3N9 Q9Y3P9 Q9Y3Q7 Q9Y3Q8 Q9Y3Y4 Q9Y421 Q9Y448 Q9Y450 Q9Y458 Q9Y463 Q9Y466 Q9Y473 Q9Y4A5 Q9Y4A9 Q9Y4B4 Q9Y4C8 Q9Y4D7 Q9Y4E6 Q9Y4G2 Q9Y4G6 Q9Y4H4 Q9Y4K1 Q9Y4K4 Q9Y4L5 Q9Y4P3 Q9Y4P8 Q9Y4P9 Q9Y4W2 Q9Y4X1 Q9Y4Z2 Q9Y508 Q9Y519 Q9Y530 Q9Y534 Q9Y535 Q9Y543 Q9Y561 Q9Y570 Q9Y573 Q9Y577 Q9Y584 Q9Y585 Q9Y587 Q9Y597 Q9Y5B9 Q9Y5C1 Q9Y5E1 Q9Y5E2 Q9Y5E4 Q9Y5F1 Q9Y5F2 Q9Y5F3 Q9Y5F6 Q9Y5F7 Q9Y5F8 Q9Y5F9 Q9Y5G0 Q9Y5G2 Q9Y5G4 Q9Y5G6 Q9Y5G7 Q9Y5G8 Q9Y5H0 Q9Y5H1 Q9Y5H3 Q9Y5H4 Q9Y5H5 Q9Y5H8 Q9Y5I0 Q9Y5I1 Q9Y5I3 Q9Y5J1 Q9Y5K1 Q9Y5K5 Q9Y5K8 Q9Y5L5 Q9Y5M8 Q9Y5P0 Q9Y5P1 Q9Y5P8 Q9Y5Q6 Q9Y5Q9 Q9Y5R2 Q9Y5R8 Q9Y5S2 Q9Y5U2 Q9Y5U9 Q9Y5V0 Q9Y5W7 Q9Y5W9 Q9Y5X2 Q9Y5Y0 Q9Y5Y3 Q9Y5Y4 Q9Y5Y6 Q9Y5Y7 Q9Y5Z7 Q9Y5Z9 Q9Y605 Q9Y608 Q9Y615 Q9Y624 Q9Y639 Q9Y651 Q9Y664 Q9Y673 Q9Y675 Q9Y679 Q9Y698 Q9Y698 Q9Y6A4 Q9Y6B2 Q9Y6B6 Q9Y6D5 Q9Y6E2 Q9Y6F8 Q9Y6G5 Q9Y6H6 Q9Y6M0 Q9Y6N5 Q9Y6N6 Q9Y6N9 Q9Y6U7*

**Table 6 genes-14-01915-t006:** The first 10 enriched pathways, in order of statistical relevance, obtained using the gene list of community 10 as input data.

	Pathway Name	*p* Value	FDR Correction	Bonferroni Correction
(1)	Olfactory Signaling Pathway	1.25E-07	1.54E-04	1.54E-04
(2)	Metabolism of proteins	1.06E-05	0.007	0.013
(3)	Post-translational protein modification	1.16E-05	0.005	0.014
(4)	Leishmania parasite growth and survival	6.71E-05	0.021	0.082
(5)	Anti-inflammatory response favouring Leishmania parasite infection	6.71E-05	0.016	0.082
(6)	Signaling by Rho GTPases, Miro GTPases and RHOBTB3	6.83E-05	0.014	0.084
(7)	Signaling by Rho GTPases	7.20E-05	0.013	0.089
(8)	TCF dependent signaling in response to WNT	9.41E-05	0.014	0.116
(9)	Signaling by WNT	1.17E-04	0.016	0.143
(10)	Transcriptional regulation by RUNX1	1.20E-04	0.015	0.148

**Table 7 genes-14-01915-t007:** The 20 multilayer genes and their affected biological pathways.

Gene Name	ID	Pathways
*CYTH3*	*O43739*	Vesicle-mediated transport; Membrane trafficking; Intra-golgi and retrograde golgi-to-ER traffic
*GSPT1*	*P15170*	Translation; Metabolism of RNA; Metabolism of proteins
*NDUFA9*	*Q16795*	Respiratory electron transport; Respiratory electron transport, ATP synthesis via chemiosmotic coupling, and heat production by uncoupling proteins; Complex I biogenesis
*ADCYAP1*	*P18509*	Signalling pathways; Metabolism of proteins
*SERPIND1*	*P05546*	Haemostasis; Metabolism of proteins; Post-translational protein modification
*SNRPA*	*P09012*	mRNA splicing; pre-mRNA splicing; Metabolism of RNA; Processing of capped intron-containing pre-mRNA
*VDAC3*	*Q9Y277*	Metabolism of proteins; Post-translational protein modification
*FDX1*	*P10109*	Diseases of metabolism; Disease
*NR5A1*	*Q13285*	Post-translational protein modification; Gene expression (Transcription); Metabolism of proteins
*MUC2*	*Q02817*	Disease; Diseases of metabolism; Defective C1GALT1C1 causes TNPS; Termination of O-glycan biosynthesis; O-linked glycosylation; O-linked glycosylation of mucins; Metabolism of proteins; Post-translational protein modification
*NDUFA7*	*O95182*	Respiratory electron transport; Respiratory electron transport, ATP synthesis via chemiosmotic coupling, and heat production by uncoupling proteins; Complex I biogenesis
*LIMK1*	*P53667*	Thrombin signalling through proteinase activated receptors (PARs); GPVI-mediated activation cascade; G alpha (12/13) signalling events; Signalling Pathways; Disease; Signalling by Rho GTPases, Miro GTPases, and RHOBTB3
*NDUFC1*	*O43677*	Respiratory electron transport; Respiratory electron transport, ATP synthesis via chemiosmotic coupling, and heat production by uncoupling proteins; Complex I biogenesis
*CPD*	*O75976*	Signalling Pathways; Metabolism of proteins; Signalling by Rho GTPases, Miro GTPases, and RHOBTB3; Signalling by Rho GTPases; RHO GTPase cycle
*PCCB*	*P05166*	Metabolism of vitamins and cofactors; Biotin transport and metabolism
*FOLR1*	*P15328*	Metabolism of proteins; Post-translational protein modification; Vesicle-mediated transport; Membrane trafficking; Asparagine N-linked glycosylation; Transport to the golgi and subsequent modification; ER-to-golgi anterograde transport; COPI-mediated anterograde transport

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
