# Peer review of "An Exploratory Application of Multilayer Networks and Pathway Analysis in Pharmacogenomics"

_genes, 2023, doi:10.3390/genes14101915_

Round 1
Reviewer 1 Report
Thank you for your work!
The authors propose using an approach to the network analysis to provide insight into the interactions between genes, drugs, and diseases. This work may be of interest to researchers of pathogenesis. After correcting cosmetic errors, the work can be published.
The comments include requirements and recommendations.
1) ‘Gml=VL, EintraL , EinterLxL, where L = f0, 1, ...lg’ (line 112): Decipher all the symbols.
2) Figure 1: Place the figure next to the first mention of it in the text (lines 100-101).
3) ‘Then, we selected the top 10 communities, i.e. the communities in comprising interlayer relations, for example gene-drug and gene-disease relations’ (lines 177-178): If it is possible to somehow designate these communities (in a figure or table), designate.
4) ‘In Figure 3’ (line 222): You referenced incorrectly. The meaning here is the description of Figure 2. Place Figure 2 next to this link.
5) Figure 3: Place the figure next to the first mention of it in the text (line 233).
6) Figure 1, Figure 2, Figure 3: Either the title of the figures is missing, or the title is formulated in such a way that it is essentially a description. The title rarely contains a verb, reformulate the names of the pictures (for example for Figure 1: ‘Figure 1. Two toy examples of a classical biological networks (a) and a multilayer network (b). …’).
7) References (line 265): Bring the elements of References to a single format.
Author Response
Manuscript ID: genes-2620179
Manuscript title: An Explorative Application of Multilayer Networks and Pathway Analysis in Pharmacogenomics
Dear Dr. Editor,
First of all, we thank the Reviewers for the time they spent on the revision of our paper.
We addressed all the suggestions of the Editor and Reviewers, and we thank them for their comments which we found helpful for refining the manuscript and correcting some remaining errors.
We carefully read the text, fixing several typos and mistakes and we asked a native English speaker to proofread the paper.
Please find attached the point-by-point list of changes made in response to the Editor’ comments and specific requests.
We applied all comments in the revised manuscript by using the Track Changes function in Latex Overleaf.
We hope that after this revision, the manuscript is now suitable for publication and we look forward to hearing from you.
Kindest regards
Marianna Milano
(on the behalf of the authors)
Reviewer 1
The authors propose using an approach to the network analysis to provide insight into the interactions between genes, drugs, and diseases. This work may be of interest to researchers of pathogenesis. After correcting cosmetic errors, the work can be published.
The comments include requirements and recommendations.
1) ‘Gml=VL, EintraL , EinterLxL, where L = f0, 1, ...lg’ (line 112): Decipher all the symbols.
Answer: We apologize for the lack of clarity. We explain each symbol.
2) Figure 1: Place the figure next to the first mention of it in the text (lines 100-101).
Answer: We thank the reviewer for pointing out this. We move the Figure 1.
3) ‘Then, we selected the top 10 communities, i.e. the communities in comprising interlayer relations, for example gene-drug and gene-disease relations’ (lines 177-178): If it is possible to somehow designate these communities (in a figure or table), designate.
We thank the reviewer for pointing out this. We add the Table.
4) ‘In Figure 3’ (line 222): You referenced incorrectly. The meaning here is the description of Figure 2. Place Figure 2 next to this link.
Answer: We apologize for the issue. We fixed this.
5) Figure 3: Place the figure next to the first mention of it in the text (line 233). Answer: We apologize for the issue. We fixed this.
6) Figure 1, Figure 2, Figure 3: Either the title of the figures is missing, or the title is formulated in such a way that it is essentially a description. The title rarely contains a verb, reformulate the names of the pictures (for example for Figure 1: ‘Figure 1. Two toy examples of a classical biological networks (a) and a multilayer network (b). ...’).
Answer: We apologize for the issue. We add the title of figures.
7) References (line 265): Bring the elements of References to a single format.
Answer: We are grateful to the Reviewer for pointing out this. We fixed the reference format.

Reviewer 2 Report
Milano et al. have undertaken a study in which they amalgamate various databases encompassing Drug-Drug Interaction (DrDrI), Disease-Disease (DD) interaction networks, Gene-Gene (GG) interaction networks, Disease-Drug Association (DDrA) networks, among others. Following the creation of a multilayer graph, the authors have conducted preliminary analyses. My primary concerns pertain to the paper's structure, the nature of the analyses performed, and the underlying hypotheses the authors aim to examine.
Concerning the manuscript's structure, it is noteworthy that there is an absence of a dedicated Materials and Methods section. Within the Results and Discussion section, there are subsections, such as the concise database descriptions, which would be more appropriately placed in a dedicated Materials and Methods section. Similarly, the statistical methods employed by the authors would benefit from a more comprehensive explanation within the Materials and Methods section. In the Results and Discussion section, it would be advisable to split it into two distinct sections, particularly given the presence of subsection 3.4 titled "Discussion.". The length of these sections relate with my second concern: in terms of the analyses conducted, it is recommended that the authors employ a broader array of statistical tools beyond community detection. Techniques such as Layer Aggregation, Node and Edge Metrics, Path Analysis, and others could offer a more comprehensive description of the multilayer graph.
Lastly, there is some ambiguity regarding the hypotheses being tested. The article appears to primarily present a descriptive analysis. While supporting the utility of multilayer graphs is valuable, it is essential that the authors emphasize what sets their approach apart from traditional methodologies. For instance, they could elucidate the significance of entities like O95182, P15328, and P09012, as well as clarify the importance of the data presented in tables 1 and 2. A more detailed interpretation of these results would enhance the article's impact.
Additionally, it is advisable for the authors to use Ensembl gene IDs or explicitly specify the system they are employing for clarity.
Author Response
Manuscript ID: genes-2620179
Manuscript title: An Explorative Application of Multilayer Networks and Pathway Analysis in Pharmacogenomics
Dear Dr. Editor,
First of all, we thank the Reviewers for the time they spent on the revision of our paper.
We addressed all the suggestions of the Editor and Reviewers, and we thank them for their comments which we found helpful for refining the manuscript and correcting some remaining errors.
We carefully read the text, fixing several typos and mistakes and we asked a native English speaker to proofread the paper.
Please find attached the point-by-point list of changes made in response to the Editor’ comments and specific requests.
We applied all comments in the revised manuscript by using the Track Changes function in Latex Overleaf.
We hope that after this revision, the manuscript is now suitable for publication and we look forward to hearing from you.
Kindest regards
Marianna Milano
(on the behalf of the authors)
Reviewer 2
Milano et al. have undertaken a study in which they amalgamate various databases encompassing Drug-Drug Interaction (DrDrI), Disease-Disease (DD) interaction networks, Gene-Gene (GG) interaction networks, Disease-Drug Association (DDrA) networks, among others. Following the creation of a multilayer graph, the authors have conducted preliminary analyses. My primary concerns pertain to the paper's structure, the nature of the analyses performed, and the underlying hypotheses the authors aim to examine.
Concerning the manuscript's structure, it is noteworthy that there is an absence of a dedicated Materials and Methods section. Within the Results and Discussion section, there are subsections, such as the concise database descriptions, which would be more appropriately placed in a dedicated Materials and Methods section. Similarly, the statistical methods employed by the authors would benefit from a more comprehensive explanation within the Materials and Methods section. In the Results and Discussion section, it would be advisable to split it into two distinct sections, particularly given the presence of subsection 3.4 titled "Discussion.".
Answer: We thank the reviewer for pointing out this. We reorganize the manuscript Sections.
The length of these sections relate with my second concern: in terms of the analyses conducted, it is recommended that the authors employ a broader array of statistical tools beyond community detection. Techniques such as Layer Aggregation, Node and Edge Metrics, Path Analysis, and others could offer a more comprehensive description of the multilayer graph.
Answer: We thank the reviewer for pointing out this. We add Tables that summarizes topological measures computed on multilayer network.
Lastly, there is some ambiguity regarding the hypotheses being tested. The article appears to primarily present a descriptive analysis. While supporting the utility of multilayer graphs is valuable, it is essential that the authors emphasize what sets their approach apart from traditional methodologies. For instance, they could elucidate the significance of entities like O95182, P15328, and P09012, as well as clarify the importance of the data presented in tables 1 and 2. A more detailed interpretation of these results would enhance the article's impact.
Answer: we thank the reviewer to highlight this lack. In the Section Discussion, we added the following clarifying sentence: “To learn more about the role of the three hub proteins, we searched the Reactome database. We used the Reactome web pathway browser and found out that the three hub proteins are a part of the Metabolism pathway. Specifically, all three proteins have an impact on the citric acid (TCA) cycle and respiratory electron transport pathway. Moreover, proteins P09012 and O95182 regulate the respiratory electron transport, ATP synthesis by chemiosmotic coupling, and heat production by uncoupling proteins pathway.”
Additionally, it is advisable for the authors to use Ensembl gene IDs or explicitly specify the system they are employing for clarity.
Answer: We thank the reviewer for pointing out this. We clarify in the Case Study subsection that we refer to NCBI Entrez Gene IDs obtained from Stanford Biomedical Network Dataset Collection (BioSNAP).
Reviewer 3 Report
Milano, et al. discuss the application of multilayer networks in pharmacogenomics. I have a few suggestions that I believe will improve the manuscript:
· It is unclear whether the manuscript is meant to be a review article or a research article, since it has aspects of both in various sections. It should be rewritten as one or the other.
· The application of multilayer networks to pharmacogenomics is interesting, but it is unclear what is meant to be demonstrated by the case study. It would be useful to show a known relationship as a proof-of-concept that this approach is valid, followed by a novel finding to illustrate that it can uncover new relationships that are worth further exploration. Also, the methods used for the case study should be described in sufficient detail to enable replication.
· Why did the authors choose to specifically discuss the link between olfactory signaling and leishmania infection on lines 213-216? This link seems somewhat tenuous as the citation refers to a paper that focused on olfactory memory in sandflies and not humans.
· Does Table 1 show the enriched pathways for community 10 only or for all top 10 communities? If only community 10, why was this particular community selected? Why was community 1 selected to include in Figure 2 and Table 2, and community 2 in Figure 3?
· For Figure 2, the colors should be changed as red lettering on green background is difficult to read.
· Table 2 should include the gene symbol and name, as most readers are not as familiar with the ID numbers listed. The authors should also indicate which database was used for the pathway analysis.
There are several grammatical errors throughout and a few areas where the text should be rephrased to improve clarity, particularly in the last few paragraphs of the introduction and in describing the results.
Author Response
Manuscript ID: genes-2620179
Manuscript title: An Explorative Application of Multilayer Networks and Pathway Analysis in Pharmacogenomics
Dear Dr. Editor,
First of all, we thank the Reviewers for the time they spent on the revision of our paper.
We addressed all the suggestions of the Editor and Reviewers, and we thank them for their comments which we found helpful for refining the manuscript and correcting some remaining errors.
We carefully read the text, fixing several typos and mistakes and we asked a native English speaker to proofread the paper.
Please find attached the point-by-point list of changes made in response to the Editor’ comments and specific requests.
We applied all comments in the revised manuscript by using the Track Changes function in Latex Overleaf.
We hope that after this revision, the manuscript is now suitable for publication and we look forward to hearing from you.
Kindest regards
Marianna Milano
(on the behalf of the authors)
Reviewer 2
Milano, et al. discuss the application of multilayer networks in pharmacogenomics. I have a few suggestions that I believe will improve the manuscript:
It is unclear whether the manuscript is meant to be a review article or a research article, since it has aspects of both in various sections. It should be rewritten as one or the other.
Answer: We apologize since we were not able to clarify this point. We insert a paragraph in Introduction in which we clarify that we present a case study consisting of the application of multilayer network formalism and pathway enrichment analysis to improve knowledge in the pharmacogenomics field.
-> Starting from these considerations, in this work we aim to present an application of network analysis in pharmacogenomics to demonstrate how network analysis methods are able to extract hidden relationships and to discover novel knowledge i.e. identifying key genes in biological pathways involved in drug response and adverse reactions.
For this aim, we built a biological multilayer network comprising genes, drugs, diseases and their associations extracted from a public database.
Then, we analyzed the multilayer network by applying a community detection algorithm, enabling the identification of essential genes from genes-diseases-drug communities.
After that, we used the identified list of genes from the communities to perform pathway enrichment analysis (PEA) to figure out the biological function affected from the selected genes. In particular, The identified genes are detached from their biological context, making it impossible to know in which biological mechanisms and functions they are involved. To understand which biological mechanisms are affected from these communities of essential genes, it is mandatory to link each gene to the opportune biological reference context by means of the pathway enrichment analysis (PEA).
PEA links genes and groups of genes to the influenced biological pathways responsible for the disease development, the adverse drug reactions as well as the different overall survival of patients treated with the same drugs. Thus, the new knowledge allows to develop new treatments more effective than the drug repositioning strategies, as well as to realize more adequate drugs reducing or even better eliminating the onset of possible adverse drug reactions.
We hope that this may clarify that the manuscript is a research article.
The application of multilayer networks to pharmacogenomics is interesting, but it is unclear what is meant to be demonstrated by the case study. It would be useful to show a known relationship as a proof-of-concept that this approach is valid, followed by a novel finding to illustrate that it can uncover new relationships that are worth further exploration. Also, the methods used for the case study should be described in sufficient detail to enable replication.
We thank the reviewer for pointing out this. We add a paragraph that explains the steps of the applied methods, and a figure that depicts all steps.
Finally, we added the following sentence highlighting the new inferred interactions. “In conclusion, the use of multilayer to represent interaction among heterogeneous data is a novel approach, especially in the field of omics. Amultilayer approach can help researchers capture more information and obtain a more accurate understanding of gene interactions.
In literature, Shang et al. in~\cite{SHANG202180} propose a multilayer network representation learning method for predicting drug-target interactions. This method integrates information from different networks, reduces noise, and learns feature vectors of drugs and targets, overcoming the challenges of integrating multiple data types and managing network noise.
Using a multilayer network to infer new relationships among genes, diseases, and drugs is at its early stage and is a continuously developing field. This limits the possibility of validating the proposed method by comparing it with existing methodologies.
Using the proposed method, we discovered potential new relationships between leishmania and different signaling pathways, results possible only through the multilayer representation. This could help researchers to identify drugs targeting specific biological functions affected by the enriched pathways. Investigating leishmania is particularly important in the context of travel medicine. Berman reviewed several aspects of diagnosis and treatment for leishmania in~\cite{berman1997human}. With our method, we could determine which drugs could contrast the damage caused by leishmania infection.”
Why did the authors choose to specifically discuss the link between olfactory signaling and leishmania infection on lines 213-216? This link seems somewhat tenuous as the citation refers to a paper that focused on olfactory memory in sandflies and not humans.
Answer: We thank the reviewer for pointing out this. We add other literature evidence reporting these evidences, adding the following sentence: “We conducted a literature search to explore the possible connections between the results of the protein enrichment analysis. According to Bhardwaj et al., ~\cite{bhardwaj2010leishmania}, leishmania alters various signaling pathways to survive, hints in line with the other enriched pathways $6, 7, 8, 9$. Additionally, Kaiser~\cite{kaiser2019druggable} found that cyclic nucleotides such as cAMP and cGMP are crucial for parasitic proliferation and regulate functions such as auditory and olfactory senses~\cite{moon1999odorants}. Also, Rho GTPases play a role in host-pathogen interaction by controlling innate and adaptive immune responses. Pathway $1$ in Table~\ref{tab:enrich} is another signaling pathway that leishmania affects, as described in~\cite{moon1999odorants}. Moreover, Schlessinger et al.~\cite{schlessinger2009wnt} explain the vital role of the mediator of Rho GTPases in the WNT signaling pathway.
Finally, Kikuchi et al.~\cite{kikuchi2006regulation} describe the regulation of WNT signaling pathways through post/translation modifications, while Li et al.~\cite{li2019runx1} provide details on the role of RUNX1 in promoting tumor metastasis by activating WNT.”
Does Table 1 show the enriched pathways for community 10 only or for all top 10 communities? If only community 10, why was this particular community selected? Why was community 1 selected to include in Figure 2 and Table 2, and community 2 in Figure 3?
Answer: We apologize since we were not able to clarify this point. We reported in Table 1 (now Table 2 ) an example of enriched pathways using the list of proteins belonging to only community 10, whereas in Table 2 (now Table 3) we link the genes with the affected pathways. Furthermore, we have reported two examples of traditional network representation of community 1 and community 2 to demonstrate that by using this type of formalism a level of information is lost such as the interaction of some genes which is instead found in the representation of the multilayer network.
For Figure 2, the colors should be changed as red lettering on green background is difficult to read.
Answer: We thank the reviewer for pointing out this. We improve the quality of the Figure 2 (now Figure 3).
Table 2 should include the gene symbol and name, as most readers are not as familiar with the ID numbers listed. The authors should also indicate which database was used for the pathway analysis.
Answer: We thank the reviewer for pointing out this. We insert the Gene Name in Table 2 and we add the pathway database in the manuscript.
There are several grammatical errors throughout and a few areas where the text should be rephrased to improve clarity, particularly in the last few paragraphs of the introduction and in describing the results.
We apologize for the issue. We corrected the typos

Round 2
Reviewer 3 Report
I appreciate the time and effort the authors took in addressing my concerns.